# Assessment of Total Antioxidant Capacity, 8-Hydroxy-2′-deoxy-guanosine, the Genetic Landscape, and Their Associations in *BCR::ABL-1*-Negative Chronic and Blast Phase Myeloproliferative Neoplasms

**DOI:** 10.3390/ijms25126652

**Published:** 2024-06-17

**Authors:** Mihnea-Alexandru Găman, Cristina Mambet, Ana Iulia Neagu, Coralia Bleotu, Petruta Gurban, Laura Necula, Anca Botezatu, Marius Ataman, Camelia Cristina Diaconu, Bogdan Octavian Ionescu, Alexandra Elena Ghiaur, Aurelia Tatic, Daniel Coriu, Amelia Maria Găman, Carmen Cristina Diaconu

**Affiliations:** 1Faculty of Medicine, “Carol Davila” University of Medicine and Pharmacy, 010221 Bucharest, Romania; mihnea-alexandru.gaman@drd.umfcd.ro (M.-A.G.); cristina.mambet@gmail.com (C.M.); drcameliadiaconu@gmail.com (C.C.D.); auratatic@yahoo.com (A.T.); daniel_coriu@yahoo.com (D.C.); 2Department of Hematology, Centre of Hematology and Bone Marrow Transplantation, Fundeni Clinical Institute, 022328 Bucharest, Romania; ionescu.bogdan44@yahoo.com (B.O.I.); ghiaur.alexandra@gmail.com (A.E.G.); 3Department of Cellular and Molecular Pathology, Stefan S. Nicolau Institute of Virology, Romanian Academy, 030304 Bucharest, Romania; neaguanaiulia@yahoo.com (A.I.N.); cbleotu@yahoo.com (C.B.); petruta_gurban@yahoo.com (P.G.); laura.necula@virology.ro (L.N.); gnanka30@yahoo.com (A.B.); marius.ataman@virology.ro (M.A.); ccdiaconu@yahoo.com (C.C.D.); 4Department of Pathophysiology, University of Medicine and Pharmacy of Craiova, 200349 Craiova, Romania; 5Clinic of Hematology, Filantropia City Hospital, 200143 Craiova, Romania

**Keywords:** myeloproliferative neoplasms, secondary acute myeloid leukemia, antioxidant capacity, 8-hydroxy-2′-deoxy-guanosine, reactive oxygen species, leukemic transformation, oxidative stress

## Abstract

Myeloproliferative neoplasms (MPNs), namely, polycythemia vera (PV), essential thrombocythemia (ET), and primary myelofibrosis (PMF), are clonal stem cell disorders defined by an excessive production of functionally mature and terminally differentiated myeloid cells. MPNs can transform into secondary acute myeloid leukemia (sAML/blast phase MPN) and are linked to alterations in the redox balance, i.e., elevated concentrations of reactive oxygen species and markers of oxidative stress (OS), and changes in antioxidant systems. We evaluated OS in 117 chronic phase MPNs and 21 sAML cases versus controls by measuring total antioxidant capacity (TAC) and 8-hydroxy-2′-deoxy-guanosine (8-OHdG) concentrations. TAC was higher in MPNs than controls (*p* = 0.03), particularly in ET (*p* = 0.04) and PMF (*p* = 0.01). *MPL W515L*-positive MPNs had higher TAC than controls (*p* = 0.002) and triple-negative MPNs (*p* = 0.01). PMF patients who had treatment expressed lower TAC than therapy-free subjects (*p* = 0.03). 8-OHdG concentrations were similar between controls and MPNs, controls and sAML, and MPNs and sAML. We noted associations between TAC and MPNs (OR = 1.82; *p* = 0.05), i.e., ET (OR = 2.36; *p* = 0.03) and PMF (OR = 2.11; *p* = 0.03), but not sAML. 8-OHdG concentrations were not associated with MPNs (OR = 1.73; *p* = 0.62) or sAML (OR = 1.89; *p* = 0.49). In conclusion, we detected redox imbalances in MPNs based on disease subtype, driver mutations, and treatment history.

## 1. Introduction

Classical myeloproliferative neoplasms (MPNs), i.e., polycythemia vera (PV), essential thrombocythemia (ET), and primary myelofibrosis (PMF), are clonal stem cell disorders defined by the excessive production of functionally mature and terminally differentiated myeloid cells as a consequence of the presence of mutually exclusive activating mutations in genes involved in the *JAK/STAT* signaling pathway, namely, the Janus Kinase 2 (*JAK2*), calreticulin (*CALR*), and thrombopoietin receptor (*MPL/TPOR*) genes, respectively [1,2,3,4]. Apart from their phenotypic mimicry, the absence of the *BCR::ABL-1* molecular marker, and their predisposition to develop primarily thrombotic events and bleeding episodes, PV and ET can progress to secondary myelofibrosis (MF), and all MPNs can undergo leukemic transformation to secondary acute myeloid leukemia (sAML or MPN blast phase) [1,2,5,6].

Molecular profiling has revealed the presence of a series of driver and clonal mutations explaining disease biology in MPNs. Genetic alterations in the *JAK2*, *CALR*, or *MPL* gene are classified as driver mutations, with in vivo investigations highlighting that when acquired, these gene abnormalities can “recapitulate” the PV, ET, or MF phenotype exhibited by humans affected by MPNs. Contrastingly, clonal mutations in non-driver genes involved in epigenetic regulation (*TET2*, *IDH1*, *IDH2*, *DNMT3A*, *ASXL1*, and *EZH2*), *ERK/MAPK* signaling (*KRAS* and *NRAS*), the regulation of *JAK* signaling (*CBL* and *SH2B3*), mRNA splicing (*SF3B1*, *SRSF2*, and *U2AF1*) or the functioning as transcriptional factors (*TP53*, *RUNX1*, and *NFE2*) do not result in the MPN phenotype when introduced in murine models; however, they alter the disease phenotype of MPNs harboring driver mutations in *JAK2*, *CALR*, or *MPL*. *JAK2* and *CALR* are involved in *JAK-STAT* signaling, whereas mutated *CALR* acts as a “rogue” chaperone protein that binds and activates *TPOR*. Driver mutations in the *JAK2* gene include *JAK2-V617F* (a point exon 14 mutation at codon 617 where the amino acid phenylalanine replaces the amino acid valine) as well as *JAK2* exon 12 mutations. *JAK2V617F* can be detected in >95% of PV and 55–60% of ET and PMF patients, and *JAK2* exon 12 affects less than 5% of subjects suffering from PV. Mutations in *CALR* result in the activation of *MPL* irrespective of thrombopoietin concentrations and have been identified in 25–30% of ET and 20–30% of PMF subjects, respectively. Similarly, mutations in *MPL* lead to aberrant megakaryopoiesis and have been noted in 5–7% of ET and 7–10% of PMF individuals. Briefly, the presence of driver gene mutations in MPNs is linked to an overproduction of mature erythrocytes, platelets, and leukocytes due to an activation of the *JAK-STAT* signaling pathway in a manner independent of cytokine concentrations [2,4,6].

Research conducted to unravel the pathogenic mechanisms involved in genomic instability, inflammatory state, and clonal expansion leading to the development of MPNs and their subsequent transformation into secondary MF and sAML has shown that oxidative stress (OS) might be a contributing factor to disease biology and evolution in the aforementioned hematological malignancies [3,7,8,9,10,11]. MPNs have been linked to alterations in the redox balance, with several investigations highlighting elevated concentrations of reactive oxygen species (ROS) and markers of OS damage, as well as changes in antioxidant systems in PV, ET, MF and sAML [3,12,13,14,15]. Interestingly, a couple of researchers have reported an elevation in the concentrations of antioxidant molecules in subjects diagnosed with MPNs, most likely as a compensatory mechanism in the presence of increased concentrations of ROS and other pro-oxidant compounds [3,12,13,14]. However, little is known about the status of redox imbalance in sAML, and most assessments involving individuals living with MPNs have only compared one disease subtype to healthy controls. Therefore, in the present study, we aimed to evaluate OS in a cohort of chronic phase MPN patients (MPNs) and subjects diagnosed with blast phase MPN (sAML) by measuring the total antioxidant capacity (TAC) as an analyte reflecting antioxidant levels and the concentrations of 8-hydroxy-2′-deoxy-guanosine (8-OHdG) as a marker of OS damage and by investigating the impact of clinical and laboratory parameters, in addition to driver mutations on TAC and 8-OHdG concentrations in MPNs and sAML.

## 2. Results

The study group consisted of 117 patients diagnosed with *BCR::ABL-1*-negative MPNs, 21 subjects with sAML (MPN blast phase), and 14 healthy subjects. The MPN cohort comprised individuals living with PV (n = 29), ET (n = 40), and MF (n = 48). MF patients were diagnosed with either pre-fibrotic, early-stage PMF (n = 12), or overtly fibrotic PMF (n = 36).

Patients with MPNs and sAML were similar in terms of age (*p* = 0.26), sex (*p* = 0.79), number of platelets (*p* = 0.47), *JAK2V617F* prevalence (*p* = 0.07), *CALR* type 1/2 mutation prevalence (*p* = 0.90), and *MPL W515L* prevalence (*p* = 0.21); however, WBC counts (*p* < 0.001), hemoglobin values (*p* = 0.03), and triple-negative cases for driver mutations (*p* < 0.001) were higher in sAML than MPNs.

Patients with PV, ET, and MF were similar in terms of age (*p* = 0.06), hemoglobin levels (*p* = 0.47), and number of platelets (*p* = 0.42); however, there was a female predisposition (*p* = 0.02), and there were notable differences in WBC counts (higher in PV than MF: *p* = 0.02 and higher in PV than prePMF: *p* = 0.03). The *JAK2V617F* mutation was the most commonly detected genetic alteration in MPNs, followed by *CALR* type 1 or type 2 mutation and *MPL W515L*, whereas the frequency of triple-negative MPN cases was estimated at 3.41%. Approximately 12% of the MPN cohort had a history of MPN treatment (hydroxycarbamide, anagrelide, or ruxolitinib) before enrolment in this study, <5% had experienced thrombotic events (stroke, acute myocardial infarction, or splanchnic vein thrombosis), and <10% displayed cardiometabolic risk factors (hypertension or type 2 diabetes mellitus). The demographic, clinical, and laboratory parameters of the MPN and sAML groups are reported in Table 1.

The demographic, clinical, and laboratory parameters of the MF subgroup (overall and based on prePMF or overt PMF subtype) are reported in Table 2.

The genetic landscape of MPNs was dominated by the presence of the *JAK2V617F* mutation, which was identified in 71.80% of the analyzed cases (n = 84). Mutations in the *CALR* gene were noted in 17.95% of the patients (n = 21), whereas *MPL W515L* was detected in 3.41% (n = 4) of the analyzed subjects. In the MPN cohort, eight patients (3.41%) were triple-negative for driver gene mutations. Regarding *CALR* mutations, *CALR* type 1 mutations were more frequent (n = 13) than *CALR* type 2 mutations (n = 8) in our study group.

Regarding the molecular alterations identified based on the MPN subtype, all the PV cases were *JAK2V617F*-positive. We did not detect any *JAK2* exon 12 mutations in our patients diagnosed with PV. Regarding ET, most cases harbored the *JAK2V617F* mutation (n = 26, i.e., 65.0%), whereas less than a quarter of ET subjects had genetic alterations in the *CALR* gene (n = 9, i.e., 22.50%). A percentage of 7.50% (n = 3) of ET patients were *MPL W515L*-positive, whereas 5.00% (n = 2) of them were triple-negative for driver mutations in the *JAK2*, *CALR*, or *MPL* gene, respectively. *CALR* type 1 mutations (n = 5) were more frequent than *CALR* type 2 mutations (n = 4) in ET in our investigation.

Genetic testing in MF revealed a similar mutation profile, with *JAK2V617F* ranking first in the list of driver mutations (n = 29, i.e., 60.42%) and *CALR* mutations detected in a quarter of the individuals living with MF (n = 12, i.e., 25.00%). One MF subject harbored the *MPL W515L* mutation (n = 1, i.e., 2.08%), whereas six patients were triple-negative for somatic driver mutations (n = 6, i.e., 12.50%).

When comparing the molecular landscape of prefibrotic MF (prePMF) and overt primary MF (PMF), *JAK2V617F* remained the most common genetic aberration in both prePMF (n = 6, i.e., 50.00%) and PMF (n = 23, i.e., 63.89%), followed by *CALR* mutations (n = 4, 33.33% for prePMF and n = 8, i.e., 22.22% for PMF). Driver mutations were absent in 16.67% (n = 2) and 11.11% (n = 4) of prePMF and PMF cases, respectively. There were no cases of prePMF with the *MPL W515L* mutation, whereas this genetic alteration was noted in one individual suffering from PMF (n = 1, i.e., 2.78%). *CALR* type 1 mutations were also more common in both prePMF (n = 3) and PMF (n = 5) than *CALR* type 2 mutations (prePMF: n = 1 and PMF: n = 3).

In terms of the sAML (n = 21) molecular landscape, *JAK2V617F* was the most common genetic alteration encountered (n = 11, i.e., 52.38%), followed by *CALR* mutations (n = 4, i.e., 19.05%). We did not detect any sAML cases harboring the *MPL W515L* mutation. In terms of triple-negative sAML cases, 28.57% of the recruited patients (n = 6) had not acquired any of the somatic mutations driving MPN biology. Of the *CALR*-mutated subjects, three harbored the *CALR* type 1 mutation, whereas only one had the *CALR* type 2 mutation.

Extensive genetic testing was performed using the NGS, WES, and/or SNP array in selected cases, particularly in MPN patients who experienced leukemic transformation. The main results, including TAC and 8-OHdG concentrations, are shown in Table 3.

The TAC levels were higher in MPNs than in controls (6.05 ± 1.11 vs. 5.46 ± 0.65, *p* = 0.03); however, there were no notable differences between healthy subjects (controls) and sAML (5.46 ± 0.65 vs. 5.99 ± 1.05, *p* = 0.10) or between MPNs and sAML (6.05 ± 1.11 vs. 5.99 ± 1.05, *p* = 0.83) (Figure 1). The TAC levels were elevated in patients with ET (6.12 ± 1.11 vs. 5.46 ± 0.65, *p* = 0.04) and MF (6.11 ± 1.23 vs. 5.46 ± 0.65, *p* = 0.02) (Figure 2), particularly in PMF (6.25 ± 1.30 vs. 5.46 ± 0.65, *p* = 0.01; Figure 3). The TAC concentrations in plasma samples were similar between controls and PV (*p* = 0.21) and prePMF (*p* = 0.52), respectively.

*MPL W515L*-positive MPNs had notably higher TAC concentrations than controls (6.74 ± 0.49 vs. 5.46 ± 0.65, *p* = 0.002) and triple-negative MPNs (6.74 ± 0.49 vs. 5.43 ± 0.82, *p* = 0.01). In particular, there were higher TAC levels in *MPL W515L*-positive ET than controls (6.90 ± 0.45 vs. 5.46 ± 0.65, *p* = 0.002) and triple-negative ET cases (6.90 ± 0.45 vs. 5.58 ± 0.39, *p* = 0.04). In addition, PMF patients who had received specific MPN treatment expressed lower TAC levels than therapy-free subjects (5.43 ± 0.99 vs. 6.50 ± 1.24, *p* = 0.03). Otherwise, age, sex, treatment history, or the presence of cardiometabolic risk factors did not impact TAC values in MPNs or sAML. No association was found between TAC concentrations and 8-OHdG, age, variant allele frequency, MF grade, WBC count, hemoglobin level, or platelet count in MPNs or sAML.

8-OHdG concentrations were similar between controls and MPNs (0.62 ± 0.12 vs. 0.97 ± 1.91, *p* = 0.59—irrespective of the subtype), controls and sAML (0.62 ± 0.12 vs. 0.92 ± 1.11, *p* = 0.67), and MPNs and sAML (0.97 ± 1.91 vs. 0.92 ± 1.11, *p* = 0.91). Age, sex, MF grade, and other laboratory parameters were not correlated with 8-OHdG in MPNs or sAML.

Data assessment by univariate regression revealed an association between TAC and MPNs (OR = 1.82; *p* = 0.05), particularly in ET (OR = 2.36; *p* = 0.03) and PMF (OR = 2.11; *p* = 0.03) but not in sAML. 8-OHdG concentrations were not associated with MPNs (OR = 1.73; *p* = 0.62) or sAML (OR = 1.89; *p* = 0.49).

The data regarding OS assessment in MPNs, sAML, and controls based on the presence of molecular testing for driver mutations are summarized in Table 4.

In terms of the impact of the molecular landscape on OS levels in MPNs and sAML, some remarks are warranted based on the results of the extensive genetic testing. Of the two cases of PV for whom targeted NGS was available, one patient did not present with additional mutations to *JAK2V617F*, whereas the other patient harbored mutations in *TP53* and *BCOR* genes, respectively. Interestingly, the PV patient with multiple genetic lesions exhibited higher TAC (+1 unit higher) and 8-OHdG concentrations. Thus, we may infer that the gain of additional mutations is associated with increased OS levels as both 8-OHdG and TAC values were increased, with the latter probably occurring as a compensatory mechanism to counteract elevated OS levels, including oxidative DNA damage measured by 8-OHdG. We observed that, in this peculiar case, TAC concentrations were notably higher than the controls, the entire MPN cohort, sAML subjects, and the PV subgroup. Moreover, 8-OHdG levels were lower than the controls, sAML cohort, MPN cohort, and PV subgroup specifically, reinforcing the hypothesis that TAC increased to reduce OS levels. We observed that the TAC values were closer to the ones detected in MF and PMF specifically; thus, this patient might be on the verge of evolution towards post-PV secondary MF, with OS playing a role in genomic instability. NGS was available for one patient with prePMF who was triple-negative for driver mutations but exhibited genetic lesions in *NF1*. His TAC and 8-OHdG levels were lower compared to the prePMF subgroup of subjects, controls, sAML cohort, and MPN cohort, with 8-OHdG concentrations similar to ET, suggesting that this individual with prePMF might have a similar phenotype to ET in terms of OS and that NF1 might have a lower contribution to OS generation in MPNs. What is more, the aforementioned PV and prePMF subjects were treatment-naïve; thus, there was no contribution of therapy to OS in these patients.

Extensive genetic testing was available in three cases of PMF. OS levels were highest in the patient who was older (68 years), had no exposure to MPN-specific medicines, and expressed the *CALR* type 1 driver mutation and associated genetic lesions in *NRAS* and *PTNP11* genes, respectively. Notably, TAC concentrations and 8-OHdG were considerably elevated compared to the controls, the entire MPN cohort, and MPN subtypes, as well as the sAML cohort, revealing a potential involvement of *NRAS* and *PTNP11* and their interaction with the *CALR* type 1 mutation in OS generation in PMF. Second, the youngest patient of the three, who had received ruxolitinib treatment, harbored mutations in *JAK2*, *TET2*, and *ASXL1*. However, his TAC and 8-OHdG concentrations were lower than the control, MPN, and sAML cohorts. Similarly, the other patient, who was 3 years older and had been previously prescribed anagrelide, also exhibited lower TAC and 8-OHdG values than the other patient groups and was positive for genetic lesions in *CALR* (*CALR* type 2) and *TET2*. Thus, in PMF, we can infer two potential separate or intricate hypotheses. On one hand, the contribution of the aforementioned genetic alterations in PMF might be lower than expected. On the other hand, since both individuals were exposed to treatment and the other subject with a notable elevation in TAC and 8-OHdG was treatment-naïve, we may infer that specific MPN therapy can reduce OS, including 8-OHdG levels, and thus, TAC concentrations normalize and do not experience a compensatory increase to counterbalance ROS generation. This might explain why these subjects had lower TAC levels versus healthy volunteers who acted as controls in our assessment.

Targeted NGS and other molecular studies were mostly available for sAML cases. With the exception of one case, where *FLT3*-*ITD* was detectable by multiplex PCR, all the other cases were negative for the presence of *AML1*-*ETO*, *CBFB*-*MYH11*, *PML*-*RARA*, or *MLL*-*AF9* gene fusion. The highest TAC level (6.82) was recorded in a patient with sAML post-PMF who harbored the *CALR* type 1 driver mutation and associated genetic abnormalities in *FLT3* and *ASXL1* genes, respectively, in addition to a treatment history of anagrelide use, whereas the lowest recorded TAC value (4.90) was noted in a subject with sAML post-PMF with the same driver mutation but additional genetic lesions in NRAS genes. Overall, *JAK2V617F*-mutated sAML cases with additional mutations displayed higher TAC concentrations (6.11, with a range of 5.44–6.65) than other sAML cases, controls, and chronic phase MPNs, whereas 8-OHdG levels were similar between the cohorts, with the exception of one case of sAML post-PMF (8-OHdG = 0.94 and TAC: 6.65) who had been previously treated with hydroxyurea and exhibited the *JAK2V617F* point mutation and other genetic lesions in *TET2*, *ASXL1*, *IDH2*, and *STAG2* genes, respectively. Nevertheless, these cased-inspired observations reinforce the hypothesis that a rise in TAC values acts as a compensatory mechanism in the setting of elevated OS levels and increased oxidative DNA damage. It is noteworthy that with the exception of one case of sAML in an individual with a history of ET, all the other cases were derived from the leukemic transformation of PMF. Thus, as our findings highlight elevated OS in PMF, these data might suggest a key role for OS in genomic instability and the evolution of chronic phase MPNs to a blast phase, i.e., sAML.

## 3. Discussion

In this study, we evaluated TAC and 8-OHdG levels in MPNs and sAML and examined the impact of demographic, clinical, and laboratory variables, as well as genetics, on the aforementioned OS-related parameters. We detected elevated TAC concentrations in ET and PMF in comparison to controls and similar TAC values between healthy volunteers and PV, prePMF, and sAML. In addition, there were no significant differences in terms of TAC values between MPN subtypes, between MPNs and sAML, as well as between MPN subtypes and sAML.

Similarly, Genovese et al. reported elevated TAC concentrations in subjects with MF, highlighting an association between TAC values and the degree of bone marrow fibrosis, the Dynamic International Prognostic Scoring System (DIPSS) score, as well as the number of CD34-positive cells in the bloodstream. In addition, they demonstrated that MF individuals who have higher concentrations of TAC display inferior survival rates and that patients with MF who harbor the *JAK2V617F* mutation have elevated TAC values compared to *CALR*-mutated cases [3]. However, we did not detect any differences in TAC between *JAK2*-mutated and *CALR*-mutated MF cases, and there was no correlation in our data between the degree of bone marrow fibrosis and TAC values. Moreover, the TAC concentrations were similar between prePMF and overt PMF. Additionally, the aforementioned study did not have a control group; thus, it was not possible to assess the difference in TAC levels between MPNs and healthy volunteers from the general population. Nevertheless, a novel finding of our paper is that MPNs with mutated *MPL W515L* exhibit higher levels of TAC than their healthy counterparts and triple-negative MPNs. Although this genetic alteration is relatively rare in MPNs and sAML compared to the prevalence of *JAK2V617F* and *CALR* mutations, as well as triple-negative MPN cases [1,15], this study sheds light on its potential role in the response to OS and/or antioxidant defense in MPNs. Other assessments have pointed out that other OS parameters are increased in PMF. For example, malondialdehyde (MDA), purine (uric acid, xanthine, and hypoxanthine), pyrimidine (uridine, uracil, and β-pseudouridine), and nitrate and nitrite concentrations were elevated and antioxidant molecules (reduced glutathione and ascorbic acid) were decreased in PMF [16]. Nevertheless, our findings highlight that the use of specific MPN therapy (hydroxycarbamide, ruxolitinib, or anagrelide) was linked to lower TAC concentrations in PMF. The literature suggests that MPN treatment options such as *JAK1*/*2* inhibitors, namely, ruxolitinib, have been associated with a reduction in OS levels. Koyuncu and collaborators revealed that after one month of ruxolitinib administration, the concentrations of several OS and nitrosative markers, e.g., thiol metabolites and ischemia-modified albumin, significantly decreased in comparison to pretreatment values, particularly in *ASXL1*-mutated and high/intermediate-2 risk according to the DIPPS-plus score PMF [17]. In contrast to the results in which previous exposure to treatment did not impact TAC levels in PV or ET, Skov et al. concluded that the recombinant interferon alpha can modulate OS in PV, ET, and PMF by upregulating and downregulating pro-oxidant genes as well as genes involved in antioxidant defense, e.g., *NRF2*, *CAT*, *SOD2*, or *TP53* [18]. However, none of our patients received recombinant interferon alpha, and thus, the effect of this pharmacological agent on TAC and 8-OHdG remains to be examined in future studies. In line with our findings, Djikic and collaborators pointed out that TAC levels estimated via the ferric-reducing ability/antioxidant power assay (FRAP) were similar between controls and PV and ET; however, PMF patients registered the highest FRAP values in the MPN cohort. Moreover, they detected elevated concentrations of MDA, protein carbonyl, and catalase (CAT), as well as reduced superoxide dismutase (SOD) levels in PMF [19]. Furthermore, the presence of the *JAK2V617F* mutation in PMF, and the heterozygous genotype in particular, was linked to the highest CAT and FRAP values and lower SOD concentrations than the controls [19]. Contrastingly, our data did not reveal a notable impact of driver mutations or triple-negativity on TAC concentrations in PMF.

In our investigation, TAC concentrations were more elevated in ET than in healthy subjects. In addition, a novel finding reported by our assessment is that the presence of the *MPL W515L* mutation in ET was linked to elevated TAC values in comparison to healthy controls and triple-negative-ET cases; the literature has mainly analyzed the impact of *JAK2V617F* and *CALR* mutations on OS levels in MPNs. Djikic and colleagues reported that ET patients exhibited similar FRAP (an assay similar to TAC) concentrations to controls and PV cases and lower values compared to PMF cases. Moreover, they discovered elevated malondialdehyde and protein carbonyl, lower superoxide dismutase and glutathione peroxidase, and similar catalase and glutathione reductase concentrations in ET than in controls, revealing a complex pro-oxidant/antioxidant imbalance in ET. In their study, ET subjects harboring the heterozygous genotype of *JAK2V617F* displayed significantly reduced superoxide dismutase and glutathione peroxidase levels compared to the controls. Moreover, the superoxide dismutase values were notably lower than the controls in *JAK2V617F*-negative ET. However, the catalase and FRAP concentrations were similar between ET and controls, as well as between ET and other MPN subtypes, irrespective of the presence of the *JAK2V617F* mutation [19]. In contrast to our findings, Durmus et al. concluded that ET is associated with a reduction in total antioxidant status. They highlighted that malondialdehyde, the total oxidative status, and the oxidative stress index were more elevated in ET than in healthy counterparts and that the administration of specific therapy lowered pro-oxidant molecule levels in ET [20]. Our results indicate the contrary, namely, that TAC concentrations were increased in patients diagnosed with ET and that the presence of *JAK2V617F* or *CALR* mutations did not influence TAC values in ET. Additionally, we demonstrated that patients who harbor the *MPL W515L* genetic alteration exhibited higher TAC levels than the controls and triple-negative ET. Thus, further research must be carried out to examine the involvement of this driver mutation in altering the redox balance in ET and other MPNs. Similarly, Iurlo and collaborators have pointed out that an increase in the levels of antioxidant molecules might signal a compensatory mechanism in response to increased OS levels in ET. According to their research, oxidized glutathione concentrations were more elevated in ET than in the controls despite similar ROS and reduced glutathione values between the controls and ET [21]. Nevertheless, other authors have reported elevated levels of ROS and reduced TAC in MPNs, e.g., PMF and post-PV/post-ET MF, particularly in correlation with disease progression [22].

Consistent with prior studies, our results, which showed elevated TAC concentrations in PMF and ET, suggest a potential compensatory increase in antioxidant levels among MPN patients. This increase may serve to counteract the elevated levels of pro-oxidant molecules and redox imbalances.

In our study, we also measured 8-OHdG concentrations in MPNs and sAML. We discovered that the levels of this compound were similar between the control group and all MPN subtypes and sAML, as well as between MPNs and sAML. In contrast to our results, Genovese et al. highlighted that 8-OHdG is elevated in MF and that MF cells have an enhanced response to oxidative damage. However, they investigated 8-OHdG in CD34+ cultured cells, whereas we analyzed 8-OHdG concentrations in plasma samples collected from MPN and sAML patients. Yet again, they emphasized that *CALR*-mutated MF cells display higher 8-OHdG levels than JAK2-mutated MF cells, whereas the mutational landscape did not impact 8-OHdG in our assessment [3]. However, it seems that the acquisition of type 1 and type 2 *CALR* mutations increases the sensitivity of in vitro models to the damage caused by OS [23]. Further investigations are needed to confirm whether these findings can also be replicated in human subjects. Several research groups have analyzed the oxidation of nucleosides in MPNs, which may reflect the oxidative damage caused to cellular DNA and RNA more clearly than the measurement of 8-OHdG. Bjørn and team evaluated the concentrations of oxidized nucleosides in urine samples and tested whether the prescription of ruxolitinib can influence the urinary concentrations of these molecules in samples collected from MF patients. However, they concluded that ruxolitinib did not influence the values of monocyte-derived hydrogen peroxide or urinary oxidized nucleosides; however, the aforementioned pharmacological agent reduced monocyte-derived superoxide generation. Interestingly, although the observation was based on a singular case, they detected an elevation in oxidized nucleosides in one patient who progressed from MF to sAML [24]. It is beyond doubt that the response to OS is altered in MPNs as transcriptomics analysis has pointed out numerous downregulated genes that contribute to antioxidant defense, particularly the *Nrf2* gene, which contributes both to the response to OS as well as the trajectory of hematopoietic stem cells from their niche [25]. Other papers, such as the one published by Sørensen and team, have shown elevated concentrations of oxidized nucleosides in MF patients, particularly in *JAK2*-mutated and *CALR*-mutated subjects. Interestingly, oxidized nucleoside concentrations were also elevated in individuals from the general population who harbored the mutations in the above-mentioned driver genes. Thus, redox imbalances may contribute to the acquisition of somatic mutations, genomic instability, and the potential development of MPNs in the general population [26].

Our findings detected an association between TAC and MPNs (OR = 1.82; *p* = 0.05), particularly in ET (OR = 2.36; *p* = 0.03) and PMF (OR = 2.11; *p* = 0.03), reinforcing the idea that derangements in the pro-oxidant/antioxidant equilibrium might trigger the onset of myeloid malignancies, e.g., MPNs, MDS, AML, and others, as previously hypothesized [27,28].

Interestingly, we did not detect differences in TAC or 8-OHdG levels in sAML versus controls or MPN patients, even though OS seems to play key contributions in AML disease biology and response to treatment [28,29]. A potential explanation would be that sAML probably received blood transfusions before the final diagnosis of sAML was achieved, and it has been demonstrated that transfusions with packed red blood cells as well as platelet concentrates can increase the concentrations of OS markers in AML [30]. In addition, the storage of blood products is linked to an elevation in OS, and the use of transfusions is correlated with an elevation in iron stores as well as iron overload, which can interfere with the crosstalk of ROS and pro-inflammatory cytokines, leading to an increase in OS in both benign and malignant hematological disorders [31,32]. Similarly, patients diagnosed with PMF are also prone to receive blood transfusions, particularly if they are prescribed drugs such as ruxolitinib or hydroxyurea, which can lead to hematological toxicities, e.g., anemia. Thus, this might explain why TAC was elevated in PMF subjects as a compensatory mechanism to combat both disease-related and treatment-related OS.

Our investigation has strengths and limitations. To our knowledge, this is the first approach to comparing TAC and 8-OHdG levels in MPNs, sAML, and healthy volunteers, as previous studies have either examined one MPN subtype or compared only MPN subtypes to each other. In particular, most recruited cases were sampled at diagnosis. Moreover, this is the first investigation to examine the contribution of *MPL W515L* driver mutation to OS in MPNs and sAML. One of the other advantages of our assessment was the fact that MPN subjects did not have a high comorbidity burden and that many were treatment-naive. The limitations of our research include the fact that we only measured TAC and 8-OHdG as indicators of OS and that we did not evaluate the urinary levels of 8-OHdG or oxidized nucleosides. However, we must point out that urine samples were not collected in the framework of the project and were thus not available in the biobank for analysis. In addition, other types of measurements, e.g., ROS quantification, require the immediate evaluation of fresh samples due to the quick degradation of ROS, and thus, this type of assessment could not be conducted using frozen plasma. However, we plan to address these limitations in future prospective studies, in which we will collect more blood components and other types of samples (e.g., urine) from MPN and sAML patients to analyze more markers of oxidative and nitrosative stress, inflammation levels, antioxidant enzymes, neutrophil extracellular traps, and other biomarkers [33,34]. In addition, as extensive genetic testing was only available in a limited number of cases, particularly in sAML patients, several observations need to be interpreted with caution and require validation in future assessments.

## 4. Materials and Methods

**Study design, setting, and participants**. In the current observational study, conducted between March 2023 and March 2024, we enrolled 117 patients diagnosed with MPNs (PV, ET, and MF), 21 patients diagnosed with sAML (blast phase MPNs), and a control group of 14 healthy volunteers who did not suffer from any disease, were non-smokers and adults, and did not take any medicines/herbal supplements. Whenever possible, we recruited MPN and sAML patients who were at diagnosis, did not suffer from associated comorbidities, were not on specific MPN treatment (hydroxyurea, anagrelide, ruxolitinib, or other treatments), were non-smokers, and did not have a history of thrombotic or hemorrhagic events. MPN and sAML patients were diagnosed according to the World Health Organization 2016 criteria [15]. Demographic, clinical, and laboratory parameters, when available, were collected from the electronic medical records of the enrolled subjects. Screening for somatic driver mutations in the *JAK2*, *CALR*, and *MPL* genes, respectively, was performed in all MPN cases. Genetic testing using next-generation sequencing (NGS), whole-exome sequencing (WES), and/or the single nucleotide polymorphism (SNP) array was conducted in selected cases. The experiments were performed in agreement with the Declaration of Helsinki in the Department of Cellular and Molecular Pathology, Stefan S. Nicolau Institute of Virology, Romanian Academy, Bucharest, Romania, and the investigation was approved by the local ethics committee (approval no. 192/06.02.2023). The patients were recruited from the Centre of Hematology and Bone Marrow Transplantation, Fundeni Clinical Institute, Bucharest, Romania, and this study was approved by the local ethics committee (reference number 40542/07.06.2021, approved on 27 May 2021). All the enrolled subjects agreed to partake in this study and signed the written informed consent. The following individuals were excluded from the investigation: subjects who did not sign the informed consent; children suffering from MPNs; adults diagnosed with *BCR::ABL-1*-positive MPNs (chronic myeloid leukemia) or other chronic MPNs, myelodysplastic syndromes, or de novo AML; patients associated with other malignancies, psychiatric illnesses, or multimorbidity or who were on medicines that could influence their oxidative stress levels; and patients who had received multiple lines of MPN treatment. The biological samples were anonymized and stored in the biobank of the Molecular Profiling of Myeloproliferative Neoplasms and Acute Myeloid Leukemia for Designing Early Diagnostic, Prognostic and Treatment Strategies (MYELOAL-EDIAPROT) project (Competitiveness Operational Programme (COP) A1.1.4. ID: P_37_798 MyeloAL-EDiaProT, Contract 149/26.10.2016, (MySMIS2014+: 106774), MyeloAL Project).

**Sample preparation and plasma collection**. Peripheral venous blood samples were collected after 10–12 h of overnight fasting via venipuncture in EDTA-coated vacutainer blood collection tubes and then centrifuged. For the assessment of oxidative stress markers, platelet-poor plasma was collected, aliquoted, and stored at −80 Celsius degrees according to standard procedures. Genomic DNA was extracted from peripheral blood granulocytes and used for the detection of somatic driver mutations in the *JAK2*, *CALR*, or *MPL* gene, respectively.

**Measurement of TAC concentrations**. TAC was evaluated based on the instructions of the manufacturer using the MAK187 TAC assay kit (Sigma-Aldrich, St. Louis, MI, USA). In brief, platelet-poor plasma samples were diluted with PBS. To detect small antioxidant molecules, a protein mask was added according to the reagent’s protocol. The principle of the assay is based on the reduction of the cupric cation (copper Cu^2+^) to the cuprous ion (copper Cu^+^), a process that requires antioxidant molecules, followed by the chelation of the cuprous ion with a colorimetric probe. Absorbance measurement was evaluated at 570 nm. Trolox (6-hydroxy-2,5,7,8-tetramethylchroman-2-carboxylic acid) served as an antioxidant standard for assembling the standard curve. Trolox equivalents were used to express the final TAC values.

**Assessment of 8-OHdG concentrations**. 8-OHdG concentrations were evaluated by an enzyme-linked immunosorbent assay (ELISA) according to the manufacturer’s instructions using the ab201734 8-hydroxy 2 deoxyguanosine ELISA Kit, ABCAM, United Kingdom. Absorbance was measured at 450 nm by a competitive ELISA that utilizes an 8-hydroxy 2-deoxyguanosine-coated plate and an HRP-conjugated antibody for detection.

**Statistical analysis**. Statistical analysis was performed using JASP version 0.18.3.0. We expressed continuous variables as mean ± standard deviation and categorical variables as percentages/numbers. Continuous variables expressing a normal distribution were analyzed using the Student’s *t*-test or ANOVA with the Bonferroni post-hoc test, whereas those with a non-normal distribution were analyzed using the Mann—Whitney U or Kruskal–Wallis test, respectively. The associations between the continuous variables were expressed as Pearson or Spearman correlation coefficients. The chi-squared test was used to detect associations between the categorical variables. Regression analysis was carried out to investigate the associations between the outcomes and exposures. *p*-values < 0.05 were considered statistically significant. Scatterplots were generated using https://scatterplot.bar/index.html (accessed on 12 June 2024).

## 5. Conclusions

According to our findings, overt PMF and ET are associated with higher TAC concentrations than healthy controls, whereas PV, prePMF, and sAML are not. The presence of the *MPL W515L* was associated with higher TAC levels in ET. Treatment-naive PMF patients had higher TAC concentrations than PMF subjects who had received specific therapy. The concentrations of 8-OHdG in plasma samples were similar between controls, MPNs, and sAML. Elevated TAC levels were associated with the diagnosis of ET and PMF.

## Figures and Tables

**Figure 1 ijms-25-06652-f001:**
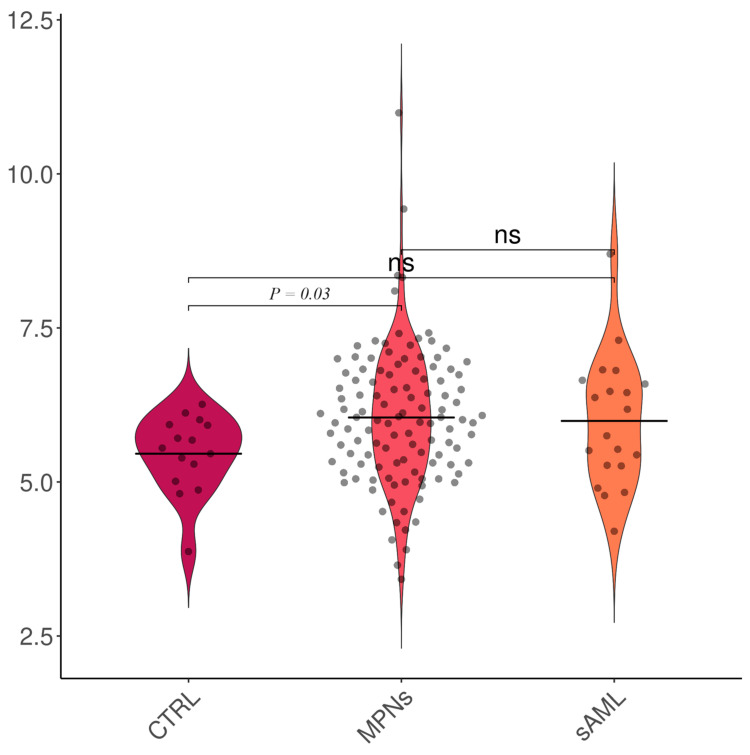
TAC evaluation in MPNs and sAML versus controls. Legend: CTRL—controls, MPNs—myeloproliferative neoplasms, sAML—secondary acute myeloid leukemia, and ns—not significant. Number of observations: CTRL (n = 14), MPNs (n = 117), and sAML (n = 21)—note that some dots may be superposed.

**Figure 2 ijms-25-06652-f002:**
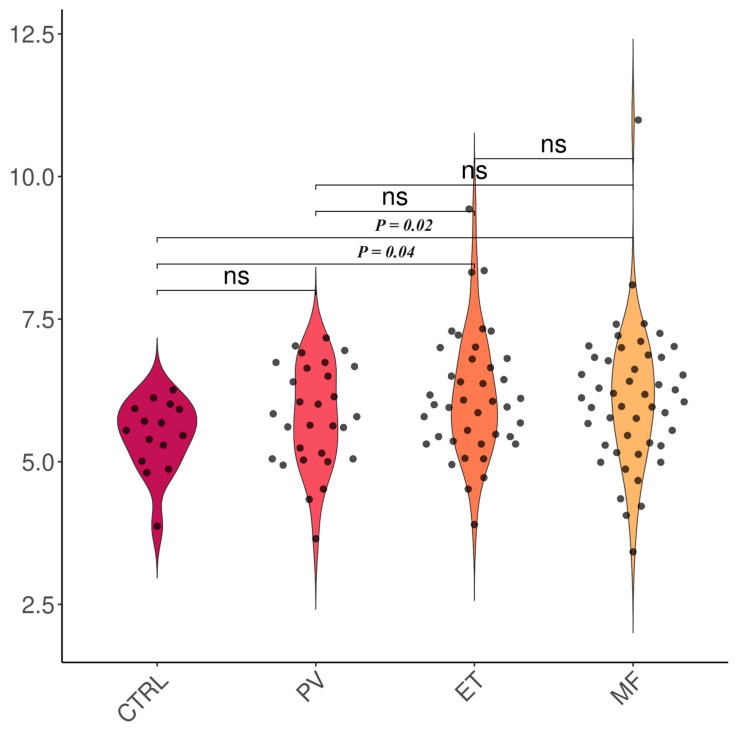
TAC evaluation in MPNs versus controls. Legend: CTRL—controls, PV—polycythemia vera, ET—essential thrombocythemia, MF—myelofibrosis, and ns—not significant. Number of observations: CTRL (n = 14), PV (n = 29), ET (n = 40), MF (n = 48)—note that some dots may be superposed.

**Figure 3 ijms-25-06652-f003:**
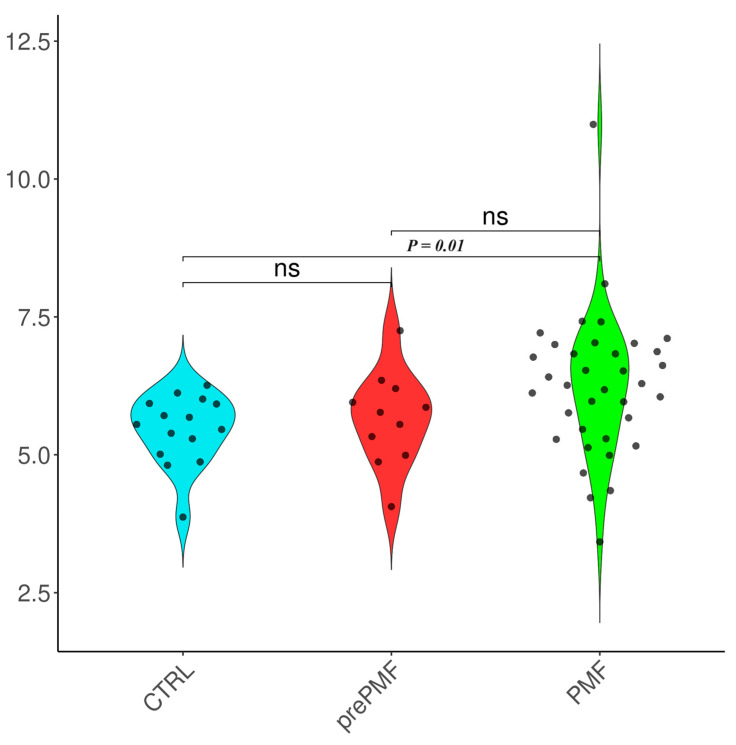
TAC evaluation in prePMF and PMF versus controls. Legend: CTRL—controls, PMF—primary myelofibrosis, prePMF—prefibrotic PMF, and ns—not significant. Number of observations: CTRL (n = 14), prePMF (n = 12), PMF (n = 36)—note that some dots may be superposed.

**Table 1 ijms-25-06652-t001:** Demographic, clinical, and laboratory parameters in MPN and sAML patients.

Characteristics	Controls	MPNs	PV	ET	MF	sAML
Patients [number]	14	117	29	40	48	21
Age [years]	55.00 ± 10.00	57.92 ± 15.31	55.10 ± 16.37	55.23 ± 16.68	61.85 ± 12.70	61.76 ± 8.77
Sex [female, %]	58.00%	58.12%	51.72%	75.00%	47.92%	47.62%
Leukocytes × 10^9^/L	4.5–9	14.23 ± 9.82	16.96 ± 16.21	14.70 ± 8.41	10.08 ± 5.96	48.88 ± 95.46
Hemoglobin [g/dL]	12–13	12.09 ± 3.51	14.83 ± 2.02	12.72 ± 3.57	11.65 ± 3.58	14.83 ± 2.02
Platelets × 10^9^/L	150–400	634.9 ± 567.0	659.3 ± 471.2	735.6 ± 725.8	545.5 ± 465.3	761.2 ± 377.7
*JAK2V617F* [%]	0.00%	71.80%	100.0%	65.00%	60.42%	52.38%
*CALR* type 1/2 [%]	0.00%	17.95%	0.00%	22.50%	25.00%	19.05%
*MPL W515L* [%]	0.00%	6.84%	0.00%	7.50%	2.08%	0.00%
Triple-negative [%]	0.00%	3.41%	0.00%	5.00%	12.50%	28.57%
No previous MPN treatment [%]	0.00%	88.03%	96.55%	90.00%	81.25%	61.90%
History of thrombosis [%]	0.00%	4.27%	5.00%	0.00%	4.17%	9.52%
Cardiometabolic risk factors [%]	0.00%	8.55%	10.34%	5.00%	8.33%	23.81%

Legend: MPNs, myeloproliferative neoplasms. sAML, secondary acute myeloid leukemia. PV, polycythemia vera. ET, essential thrombocythemia. MF, myelofibrosis.

**Table 2 ijms-25-06652-t002:** Demographic, clinical, and laboratory parameters in MF patients.

Characteristics	MF (prePMF + PMF)	prePMF	PMF
Patients [number]	48	12	36
Age [years]	61.85 ± 12.70	54.50 ± 15.80	64.31 ± 10.65
Sex [female, %]	47.92%	50.00%	47.22%
Leukocytes × 10^9^/L	10.08 ± 5.96	8.15 ± 3.99	10.75 ± 6.42
Hemoglobin [g/dL]	11.65 ± 3.58	10.76 ± 2.84	11.95 ± 3.80
Platelets × 10^9^/L	545.5 ± 465.3	639.1 ± 609.5	512.3 ± 409.5
*JAK2V617F* [%]	60.42%	50.00%	63.89%
*CALR* type 1/2 [%]	25.00%	33.33%	22.22%
*MPL W515L* [%]	2.08%	0.00%	2.78%
Triple-negative [%]	12.50%	16.67%	11.11%
No previous MPN treatment [%]	81.25%	91.67%	77.78%
History of thrombosis [%]	4.17%	8.33%	2.77%
Cardiometabolic risk factors [%]	8.33%	25.00%	2.78%

Legend: MPN—myeloproliferative neoplasm, MF—myelofibrosis, prePMF—prefibrotic primary myelofibrosis, and PMF—primary myelofibrosis.

**Table 3 ijms-25-06652-t003:** Assessment of the relationship between MPN chronic/blast phase diagnosis, OS levels, and molecular studies.

Diagnosis	Obs.	OS Evaluation	Main Results of Genetic Testing
PV	33 years Male Previous treatment: none	TAC 6.14 8-OHdG 0.57	Driver mutation: *JAK2 V617F* 88% *TP53* c.388C>T, p.(L130F), 35%, loss of function, COSM11449 *BCOR* c.4938_4939delCT, p.(L1647fs*4), 42%, loss of function, likely pathogenic, unreported
PV	65 years Female Previous treatment: none	TAC 5.15 8-OHdG 0.51	Driver mutation: *JAK2 V617F* Targeted NGS: negative for other mutations
prePMF	26 years Female Previous treatment: none	TAC 4.87 8-OHdG 0.58	Driver mutation: triple-negative *NF1* c.1792A>C (K598Q), 30%, missense, probably damaging
PMF	48 years Female Previous treatment: RUX	TAC 5.29 8-OHdG 0.58	Driver mutation: *JAK2 V617F* 92.8% *TET2* c.593dupT, 33.5%, rs748109142, unknown significance *ASXL1* c.1934dupG (p.G646Wfs*12), 32.1%, COSM1411076
PMF	51 years Female Previous treatment: ANA	TAC 4.99 8-OHdG 0.50	Driver mutation: *CALR* type 2 *CALR* c.1154_1155ins TTGTC 45.2% *TET2* c.4354C>T p.R1452* (nonsense) 21% COSM41706
PMF	68 years Female Previous treatment: none	TAC 7.00 8-OHdG 16.35	Driver mutation: *CALR* type 1 *NRAS* c.190T>G (p.Y64D) 44.8%, COSM1666991 *PTNP11* c.1516T>A (p.S502T) 21.4%, COSM14258
sAML Provenance: PMF	72 years Female Previous treatment: HU	TAC 6.65 8-OHdG 0.94	Driver mutation: *JAK2 V617F* (DNA from CD34+ cells: 6% and DNA from CD3+ cells: 2.5%) *TET2* c.1648C>T, p.R550*, COSM41644 (DNA from CD34+ cells: 48.6% and DNA from CD3+ cells: 37.8%) *ASXL1* c.2066C>G, p.S689*, COSM133037 (DNA from CD34+ cells: 43% and DNA from CD3+ cells: 35.4%) *STAG2* c.1840C>T, p.R614*, COSM166815 (DNA from CD34+ cells: 40.9% and DNA from CD3+ cells: 33.9%) *TET2* c.5611_5618delATTCTCAT, p.I1871Ter, unreported (DNA from CD34+ cells: 36.8% and DNA from CD3+ cells: 36.5%) *IDH2*, c.419G>A, p.R140Q, COSM41590 (DNA from CD34+ cells: 27.2% and DNA from CD3+ cells: 19.70%)
sAML Provenance: ET	76 years Female Previous treatment: HU	TAC 5.44 8-OHdG 0.52	Driver mutation: *JAK2 V617F* 78.9% *DNMT3A* c.2322 + 1G>A and p.? splice-site mutation pathogenic 43%
sAML Provenance: PMF	51 years Female Previous treatment: AZA	TAC 6.47 8-OHdG 0.52	Driver mutation: *JAK2 V617F* (DNA from CD34+ cells: 72% and DNA from CD3+ cells: 10%) *TP53* c.537T>A, p.179Q, missense, COSV52669519 (DNA from CD34+ cells: 100% and DNA from CD3+ cells: 35%) *SRSF2* c.284C>T, p.P95L, missense, COSV57969830 (DNA from CD34+ cells: 49% and DNA from CD3+ cells: 12%)
sAML Provenance: PMF	54 years Female Previous treatment: ANA	TAC 6.82 8-OHdG 0.52	Driver mutations: *CALR* type 1 *FLT3* 57.2% (unknown clinical significance and extremely rare in the general population) *ASXL1* c.2476_2485 dupGGAACTGGCC 28.2%, unreported
sAML Provenance: PMF	56 years Male Previous treatment: HU	TAC 5.75 8-OHdG 0.49	Driver mutation: triple-negative *ASXL1* c.1888_1910del23 (p.E635fs*15) 73.7% COSM36165 *TP53* c.395A>G (p.K132R) 66.3% COSM308311 *EZH2* c.1979G>A (p.G660E) 39.5% (unreported in COSMIC and deleterious/probably damaging)
sAML Provenance: PMF	50 years Female Previous treatment: RUX	TAC 5.53 8-OHdG 0.51	Driver mutation: *JAK2 V617F* *TET2* c.1088C>T (p.P363L) 48.4%, and COSM5020142 (germline and clinical significance not provided) *KIT* c.1621A>C (p.M541L), 52.1%, COSM28026
sAML Provenance: PMF	69 years Female Previous treatment: none	TAC 4.90 8-OHdG 0.51	Driver mutation: *CALR* type 1 51% *NRAS* c.190T>G, p.Y64D, (DNA from CD34+ cells: 90.9% and DNA from CD3+ cells: 49%—germline) *NRAS* c.35G>A, p.G12D, 43%, COSM564 SNP array: del7q22.1, del8q11.1-q11.21, del10p12.1-p11.22, del11p14.1-p11.2, and delXp11.4 UPD1p
sAML Provenance: PMF	63 years Male Previous treatment: AZA	TAC 6.45 8-OHdG 0.51	Driver mutation: *JAK2 V617F* (DNA from CD34+ cells: 95.1% and DNA from CD3+ cells: 17.8%) *RUNX1* c.364_365insAA, frameshift, p.Gly122fsTer12 (DNA from CD34+ cells: 60.9% and DNA from CD3+ cells: 14.8%) *CSF3R* c.2492C>T p.A831V (unreported) (DNA from CD34+ cells: 47.4%) *IDH1* c.395G>A, p.R132H, COSM28746 (DNA from CD34+ cells: 34.4% and DNA from CD3+ cells: 5.00%) *PHF6* c.385C>T, p.R129*, COSM4606367 (DNA from CD34+ cells: 6.00%)
sAML Provenance: PMF	57 years Female Previous treatment: none	TAC 6.37 8-OHdG 0.52	Driver mutation: triple-negative sAML: *FLT3-ITD*
sAML Provenance: PMF	57 years Male Previous treatment: none	TAC 5.27 8-OHdG 0.52	Driver mutation: *CALR* type 1 58.1% SNP array: *UPD11q* Targeted NGS: DNA from PBMC *ASXL1* c.1773C>A, p.Y591*, 49.3%, COSM1681609 *CBL* c.1111T>A, p.Y371N, 93.1%, COSM5031014 DNA from granulocytes *ASXL1* c.1773C>A, p.Y591*, 43.9%, COSM1681609 *CBL* c.1111T>A, p.Y371N, 98%, COSM5031014 WES: DNA from PBMC *CALR* del52 39% *ASXL1* c.1773C>A, p.Y591*, 49%, COSM1681609 *CBL* c.1111T>A (p.Y371N), 96%, COSM5031014 *NBN* c.511A>G, p.I171V, missense, unknown significance, 55% STAT5A c.2118dupT, p.V707fs, 56%

Legend: TAC—total antioxidant capacity, 8-OHdG—8-hydroxy-2′-deoxy-guanosine, MPNs—myeloproliferative neoplasms, sAML—secondary acute myeloid leukemia, PV—polycythemia vera, ET—essential thrombocythemia, MF—myelofibrosis, prePMF—prefibrotic primary myelofibrosis, PMF—primary myelofibrosis, Obs.—Observation(s), OS—oxidative stress, HU—hydroxyurea, AZA—azacytidine, RUX—ruxolitinib, ANA—anagrelide, NGS—next-generation sequencing, WES—whole-exome sequencing, PBMC—peripheral blood mononuclear cells, DNA—deoxyribonucleic acid, and SNP—single nucleotide polymorphism.

**Table 4 ijms-25-06652-t004:** Assessment of OS based on MPN/sAML diagnosis and molecular testing for mutations in driver genes.

Subgroup	TAC [Trolox Equivalents]	8-OHdG [ng/mL]
** *Controls* **	5.46 ± 0.65	0.62 ± 0.12
** *MPNs (entire cohort)* **	6.05 ± 1.11	0.97 ± 1.91
** *sAML* **	5.99 ± 1.05	0.92 ± 1.11
** *PV* **	5.79 ± 0.89	0.81 ± 0.62
*JAK2V617F* (+)	5.79 ± 0.89	0.81 ± 0.62
** *ET* **	6.12 ± 1.11	0.56 ± 0.09
*JAK2V617F* (+)	6.15 ± 1.25	0.54 ± 0.10
*CALR* (+)	6.04 ± 0.77	0.58 ± 0.10
*MPL W515L* (+)	6.90 ± 0.45	0.58 ± 0.09
Triple-negative	5.58 ± 0.38	0.57 ± 0.08
** *MF* **	6.11 ± 1.23	1.27 ± 2.94
*JAK2V617F* (+)	6.38 ± 1.35	0.79 ± 0.61
*CALR* (+)	5.80 ± 0.87	2.45 ± 5.24
*MPL W515L* (+)	6.26 ± 0.00	0.58 ± 0.00
Triple-negative	5.37 ± 0.94	0.58 ± 0.11
** *prePMF* **	5.65 ± 0.84	0.71 ± 0.47
*JAK2V617F* (+)	5.68 ± 0.92	0.64 ± 0.21
*CALR* (+)	5.89 ± 0.96	0.91 ± 0.82
*MPL W515L* (+)	-	-
Triple-negative	5.10 ± 0.32	0.54 ± 0.05
** *(overt) PMF* **	6.25 ± 1.30	1.66 ± 3.82
*JAK2V617F* (+)	6.54 ± 1.40	0.89 ± 0.78
*CALR* (+)	5.76 ± 0.90	3.67 ± 7.08
*MPL W515L* (+)	6.26 ± 0.00	0.58 ± 0.00
Triple-negative	5.51 ± 1.17	0.62 ± 0.17

Legend: TAC—total antioxidant capacity, 8-OHdG—8-hydroxy-2′-deoxy-guanosine, MPNs—myeloproliferative neoplasms, sAML—secondary acute myeloid leukemia, PV—polycythemia vera, ET—essential thrombocythemia, MF—myelofibrosis, prePMF—prefibrotic primary myelofibrosis, and PMF—primary myelofibrosis.

## Data Availability

The original contributions presented in this study are included in this article. Further inquiries can be directed to the corresponding author.

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
