# Peer review of "Assessment of Total Antioxidant Capacity, 8-Hydroxy-2′-deoxy-guanosine, the Genetic Landscape, and Their Associations in BCR::ABL-1-Negative Chronic and Blast Phase Myeloproliferative Neoplasms"

_ijms, 2024, doi:10.3390/ijms25126652_

Round 1

Reviewer 1 Report

Comments and Suggestions for Authors

Comments to the Author

In the manuscript entitled “Assessment of total antioxidant capacity and 8-hydroxy-2′-de-2 oxy-guanosine levels in BCR::ABL-1 negative chronic and blast 3 phase myeloproliferative neoplasms”, the authors present a study related to markers of oxidative stress (OS) in a cohort of chronic phase MPN (myeloproliferative neoplasms) patients, including ET, PV and PMF (classical MPNs), and subjects diagnosed with sAML (secondary acute myeloid leukemia). They measured the total antioxidant capacity (TAC) as an analyte reflecting anti-oxidant levels and the concentrations of 8-hydroxy-2′-deoxy-guanosine (8-OHdG) as a marker of OS damage, and they estimated the relationship of clinical and laboratory parameters, in addition to driver mutations, with TAC and 8-OHdG concentrations in MPNs and sAML.

 Overall, the paper is well-written. The study is of interest, however, this reviewer feels that the paper could be significantly improved.

The major drawback of this study is the lack of more specific tests and more detailed analysis of both antioxidant capacity and markers of  oxidative stress damage, to support the conclusions.

The authors themselves presented the advantages and disadvantages of their study, in the conclusion:

“Our investigation has strengths and limitations. To our knowledge, this is the first approach at comparing TAC and 8-OHdG levels in MPNs, sAML, and healthy volunteers, as previous studies have either examined one MPN subtype or compared only MPN sub-types with each other. In particular, most recruited cases were sampled at diagnosis. Moreover, this is the first investigation to examine the contribution of MPL W515L driver mutation to OS in MPNs and sAML. Other advantages of our assessment were the fact that MPN subjects did not have a high comorbidity burden and that many were treatment-naive. Limitations of our research include the fact that we only measured two parameters related to OS or that we did not evaluate urinary levels of 8-OHdG or oxidized nucleosides. However, this patient was young (26 years) as opposed to other subjects in whom aging also could have played a notable contribution to OS.”

 Last sentence:

 “However, this patient was young (26 years) as opposed to other subjects in whom aging also could have played a notable contribution to OS”

is unclear.

Author Response

Dear Editor-in-Chief of International Journal of Molecular Sciences,

Dear Academic Editor,

Dear Peer-Reviewers,

We are thankful for your valuable comments and constructive criticism regarding our paper. We have performed all requested revisions and all changes have been explained and/or highlighted in yellow in the text. We do hope that in its revised form the manuscript warrants publication in the International Journal of Molecular Sciences.

REVIEWER 1 - Comments to the Author

In the manuscript entitled “Assessment of total antioxidant capacity and 8-hydroxy-2′-de-2 oxy-guanosine levels in BCR::ABL-1 negative chronic and blast 3 phase myeloproliferative neoplasms”, the authors present a study related to markers of oxidative stress (OS) in a cohort of chronic phase MPN (myeloproliferative neoplasms) patients, including ET, PV and PMF (classical MPNs), and subjects diagnosed with sAML (secondary acute myeloid leukemia). They measured the total antioxidant capacity (TAC) as an analyte reflecting anti-oxidant levels and the concentrations of 8-hydroxy-2′-deoxy-guanosine (8-OHdG) as a marker of OS damage, and they estimated the relationship of clinical and laboratory parameters, in addition to driver mutations, with TAC and 8-OHdG concentrations in MPNs and sAML.

Overall, the paper is well-written. The study is of interest, however, this reviewer feels that the paper could be significantly improved.

The major drawback of this study is the lack of more specific tests and more detailed analysis of both antioxidant capacity and markers of  oxidative stress damage, to support the conclusions.

The authors themselves presented the advantages and disadvantages of their study, in the conclusion:

“Our investigation has strengths and limitations. To our knowledge, this is the first approach at comparing TAC and 8-OHdG levels in MPNs, sAML, and healthy volunteers, as previous studies have either examined one MPN subtype or compared only MPN sub-types with each other. In particular, most recruited cases were sampled at diagnosis. Moreover, this is the first investigation to examine the contribution of MPL W515L driver mutation to OS in MPNs and sAML. Other advantages of our assessment were the fact that MPN subjects did not have a high comorbidity burden and that many were treatment-naive. Limitations of our research include the fact that we only measured two parameters related to OS or that we did not evaluate urinary levels of 8-OHdG or oxidized nucleosides. However, this patient was young (26 years) as opposed to other subjects in whom aging also could have played a notable contribution to OS.”

Response: Thank you for your valuable feedback and constructive criticism. In this investigation, we used plasma samples to evaluate oxidative stress levels in MPNs and sAML and thus we were limited by the types of oxidative stress markers that could be measured in plasma. Moreover, the plasma samples collected were limited in quantity. Frozen plasma samples are not valid to be used in the measurement of reactive oxygen species, for examples, as ROS are quickly degraded. Moreover, we could not assess urinary levels of 8OHdG as urine samples were not collected from the patients recruited for the MYELOAL-EDIAPROT project. However, we will try to conduct a prospective investigation in the near future in which we will collect multiple types of samples from MPNs and sAML patients and quantify more markers of oxidative and nitrosative stress, as well as pro-inflammatory cytokines, as instructed and as evidenced by the literature – see newly added references. In addition, one must take into consideration that we were limited by the funding of the project.

Last sentence:  “However, this patient was young (26 years) as opposed to other subjects in whom aging also could have played a notable contribution to OS” is unclear.

Response: Thank you for your observation. We have deleted this phrase, it was inserted there by error.

Reviewer 2 Report

Comments and Suggestions for Authors

A detailed description of the genetic mutations of the different MPN entities and AML related to TAC and 8OHdG and the applied therapy is given, but I think that the sample of three PMF patients is not large enough to make a hypothesis and draw a conclusion.

Author Response

Dear Editor-in-Chief of the International Journal of Molecular Sciences,

Dear Academic Editor,

Dear Peer-Reviewer,

We are thankful for your valuable comments and constructive criticism regarding our paper. We have performed all requested revisions and all changes have been explained and/or highlighted in yellow in the text. We do hope that in its revised form the manuscript warrants publication in the International Journal of Molecular Sciences.

REVIEWER 2 - Comments and Suggestions for Authors

A detailed description of the genetic mutations of the different MPN entities and AML related to TAC and 8OHdG and the applied therapy is given, but I think that the sample of three PMF patients is not large enough to make a hypothesis and draw a conclusion.

Response: Thank you for your valuable feedback and constructive criticism. We have mentioned in the limitations of our investigation that extensive genetic testing was not available in all cases of PMF and thus several of our remarks were simple observations that require validation in future studies.

Reviewer 3 Report

Comments and Suggestions for Authors

Authors reported major genetic mutations in 117 myeloproliferative neoplasm (MPN) and 21 secondary acute myeloid leukemia (sAML) patients and analyzed levels of total antioxidant capacity (TAC) and 8-hydroxy-2’-deoxy-guanosine (8-OHdG) in these patients. Authors may wish to make specific revisions to further improve the manuscript. 

Specific comments:

The title of the manuscript should be revised to reflect what is reported in the manuscript since a major portion of the context were about genetic mutations (high frequencies of  JAK2V617F mutation in MPN and sAML patients along with mutations in other genes such as calreticulin (CALR) and thrombopoietin receptor (MPL)). This part should be reflected in the title.

Too many abbreviations were used in the manuscript. Authors made a good effort to spell out full term for some of the abbreviations. Still, some abbreviations were used without clear description. Authors should check through the manuscript to reduce the number of abbreviations used in the manuscript, and to spell out full term of an abbreviation at its first appearance.

Having Table 1 is a good idea. However this Table should be further improved: 1) add a  healthy control column; 2) make it clear that PV, ET and MF are part of MPN while pre-PMF and PMF are part of MF, as current listing is confusing; 3) report WBC and PLT as 109/L.  

The detection of genetic mutations used different number of samples. Authors made a good effort to state clearly how many samples was used for the analyses in each case, as described between lines 118 and 148. Since not 100% samples were used for genetic analyses, authors should provide some background information as how samples were chosen for each mutation analyses (random?) in each patient subset.

Current Figures 1 and 2 should be combined, expanded and completely reconstructed. The box plots and bell curves are redundant and unnecessary. Instead, mean and standard error ranges can be added to the dot plots. Figure 1 should include multiple panels all using TAC as Y axis. For example, the first panel should have CTRL, MPN and sAML as X axis while the second panel could have CTRL, PV, ET, MF as X axis. The third panel could have CTRL, prePMF and PMF as X axis. Authors could add fourth and more panels to describe TAC data on various patient subsets. It is also very important to provide a meaningful figure legend stating the main procedures and the number of observations used for each group.

Comments on the Quality of English Language

some English editing will help to improve the manuscript. 

Author Response

Dear Editor-in-Chief of International Journal of Molecular Sciences,

Dear Academic Editor,

Dear Peer-Reviewers,

We are thankful for your valuable comments and constructive criticism regarding our paper. We have performed all requested revisions and all changes have been explained and/or highlighted in yellow in the text. We do hope that in its revised form the manuscript warrants publication in the International Journal of Molecular Sciences.

REVIEWER 3 - Comments and Suggestions for Authors

Authors reported major genetic mutations in 117 myeloproliferative neoplasm (MPN) and 21 secondary acute myeloid leukemia (sAML) patients and analyzed levels of total antioxidant capacity (TAC) and 8-hydroxy-2’-deoxy-guanosine (8-OHdG) in these patients. Authors may wish to make specific revisions to further improve the manuscript. 

Response: Thank you for your valuable comments and constructive criticism.

Specific comments:

The title of the manuscript should be revised to reflect what is reported in the manuscript since a major portion of the context were about genetic mutations (high frequencies of  JAK2V617F mutation in MPN and sAML patients along with mutations in other genes such as calreticulin (CALR) and thrombopoietin receptor (MPL)). This part should be reflected in the title.

Response: Thank you for your valuable comment. The title has been changed to reflect that we also assessed the impact of the genetic landscape on oxidative stress in MPNs.

Too many abbreviations were used in the manuscript. Authors made a good effort to spell out full term for some of the abbreviations. Still, some abbreviations were used without clear description. Authors should check through the manuscript to reduce the number of abbreviations used in the manuscript, and to spell out full term of an abbreviation at its first appearance.

Response: Thank you for your valuable comment. We reduced the number of abbreviations, added legends below the tables to explain potential abbreviations and also added a list of abbreviations at the end of the paper.

Having Table 1 is a good idea. However this Table should be further improved: 1) add a  healthy control column; 2) make it clear that PV, ET and MF are part of MPN while pre-PMF and PMF are part of MF, as current listing is confusing; 3) report WBC and PLT as 109/L. 

Response: Thank you for your valuable comment. We split Table 1 into Table 1 which presents controls + MPNs + PV + ET + MF and Table 2 for MF + prePMF + PMF so that no confusions will arise. We also added the healthy control column and reported WBCs and PLT as instructed.

The detection of genetic mutations used different number of samples. Authors made a good effort to state clearly how many samples was used for the analyses in each case, as described between lines 118 and 148. Since not 100% samples were used for genetic analyses, authors should provide some background information as how samples were chosen for each mutation analyses (random?) in each patient subset.

Response: Thank you for your valuable comment. Somatic mutations in driver genes (JAK2, CALR, MPL) were assessed in all MPN and sAML cases. For oxidative stress evaluation, we selected samples from the biobank of the MYELOAL-EDIAPROT project based on several criteria: samples collected at the diagnosis of MPNs/sAML, samples from treatment-naive patients (as therapy also influences oxidative stress levels), samples collected preferably from younger patients (as aging also increases oxidative stress levels) and from patients without comorbidities (cardiovascular disease and other chronic comorbidities are associated with increased oxidative stress levels) etc. Extensive genetic testing was performed primarily in sAML cases and in MPN cases when samples from the chronic phase and the blastic phase of the same patient was available.

Current Figures 1 and 2 should be combined, expanded and completely reconstructed. The box plots and bell curves are redundant and unnecessary. Instead, mean and standard error ranges can be added to the dot plots. Figure 1 should include multiple panels all using TAC as Y axis. For example, the first panel should have CTRL, MPN and sAML as X axis while the second panel could have CTRL, PV, ET, MF as X axis. The third panel could have CTRL, prePMF and PMF as X axis. Authors could add fourth and more panels to describe TAC data on various patient subsets. It is also very important to provide a meaningful figure legend stating the main procedures and the number of observations used for each group.

Response: Thank you for your valuable comment. We have reconstructed the figures: Figure 1 – comparative evaluation of TAC in controls, MPNs and sAML; Figure 2 – comparative evaluation of TAC in controls, PV, ET, MF; Figure 3 – comparative evaluation of TAC in controls, prePMF, PMF. Number of observations were also added as well as legends for the figures.

Comments on the Quality of English Language

some English editing will help to improve the manuscript

Response: Thank you for your valuable comment. The English language and scientific writing were polished by a native English speaker.

Round 2

Reviewer 1 Report

Comments and Suggestions for Authors

Overall, the paper is well-written, and the study is of interest. After revision, the authors have significantly improved the quality of the manuscript. They changed the title, tables and figures accordingly. They adequately responded to the major drawback of this study (the lack of more specific tests and more detailed analysis of both antioxidant capacity and markers of oxidative stress damage). Considering their great effort and competence, I believe that the paper can be published.